# Machine Learning and Rules Induction in Support of Analog Amplifier Design

**Malinka Ivanova** [1],*[ID] **and Miona Andrejević Stošović** [2]

1    Department of Informatics, Faculty of Applied Mathematics and Informatics, Technical University of Sofia, Sofia 1797, Bulgaria
2    Department of Electronics, Faculty of Electronic Engineering, University of Niš, Niš 18000, Serbia
*    Correspondence: m_ivanova@tu-sofia.bg

**Abstract:** The aim of the paper is to present a two-step method for facilitating the design of analog amplifiers taking into account the bottom–top approach and utilizing machine learning techniques. The X-chart and a framework describing the specificity of analog circuit design using machine learning are introduced. The possibility of libraries with open machine learning models to support the designer is also discussed. The proposed method is verified for a three-stage amplifier design. In the first step, the stage type is predicted with 89.74% accuracy as the applied learner is a Decision Tree machine learning algorithm. Moreover, two induction rule algorithms are used for predictive logic generation. In the second step, some typical parameters for a given stage are predicted considering four learners: Decision Tree, Random Forest, Gradient Boosted Trees, and Support Vector Machine. The most suitable is found to be Support Vector Machine, which is characterized with the smallest obtained errors.

**Keywords:** analog design; machine learning; amplifier circuits; X-chart; framework for analog circuits design

## 1. Introduction

Amplifier design is a complex problem related to finding the best circuit structure according to specifications predefined by the user. This specification explains the desired function and application of the designed circuit, and its required electrical parameters and characteristics. To solve this design task, the designer should know and understand the principle of operation and specific features of a wide variety of simple circuits, in addition to the methods for designing complex electronic modules and devices. In analog electronics, some simple building stages serve as the basis of construction of different analog circuits such as amplifiers, functional converters, filters, and generators [1,2]. Thus, knowledge about circuit operability and possible circuit variants, and about the theory of how to construct a circuit according to a given specification, are a very important part of the design process. The circuit design is also supported by Electronic Design Automation (EDA) software equipped with multiple component libraries and appropriate instrumentation. This saves time, resources, and effort for designers. Some errors and non-suitable circuit variants can be avoided. Recently, machine and deep learning have been utilized in an assistive role in circuit design to automate engineering tasks [3–5]. Machine learning-based approaches to design rely on the collected data, a strong understanding of the theory in electronics, and the practically proven methods. This knowledge should be combined with familiarity regarding the specificity and advantages of machine learning algorithms [6–8] that should support the right design decisions. Furthermore, Hamolia and Melnyk show the need for new methodologies for high-level automated design, integrated in EDA software, which is driven by continuous technology development [9]. The authors point out the appearance of a new scientific field related to machine learning-based EDA for facilitating all phases

of chip design. Ren et al. group the applications of machine learning for solving EDA problems to predictors, optimizers, and generators [10]. They argue that conventional EDA algorithms should collaborate with machine learning for achieving greater efficiency. It seems that the design process could be facilitated in both directions: from top to bottom and from bottom to top, and at behavioral, structural, and physical levels through machine learning models. Some research papers address similar topics, showing positive results, successful implementations, and challenging issues. Dieste-Velasco et al. present a methodology for improving the design of electronic circuits, driven by artificial neural network algorithms and the statistical technique design of experiments [11]. They conclude that the proposed approach can be used for efficient behavioral modeling of electronic circuits and for the prediction of some parameters. Guerra-Gomez et al. investigate the speed of regression techniques used in the design of medium- and large-scale electronic circuits and prove the suitability of regression algorithms for circuit modeling with high speed and high accuracy [12]. The research team of Hasani et al. propose a compositional method for building an artificial neural network used for modeling complex analog integrated circuits and reduced simulation time is demonstrated [13]. Mina et al. summarize the existing scientific achievements in automating the design process of integrated analog circuits (on MOS, CMOS technology), pointing out the advantages of machine learning techniques (supervised, unsupervised, reinforcement learning) for circuit designers [14].

Machine learning utilization at the physical level of chip design is also discussed in several scientific publications, which describe the current progress and bottlenecks of component placement and routing [15–17]. Time saving for optimal component placement on the printed circuit board (PCB), avoiding concurrency issues in routing, and increasing the designer's efficacy are among the future problems that should be solved, including through usage of machine learning.

Obviously, the evolution of machine learning and data science has led to inventing novel methodological solutions in electronics and circuit design, as indicated by the increased scientific interest in recent years. The reported findings are related to design optimization [18], object detection [19], defect identification [20], classification [21], etc.

The aim of the paper is to present a method for facilitating the design of analog amplifiers based on utilization of machine learning algorithms following the bottom–top design strategy. The X-chart and a framework reflecting on the specificity of analog circuit design using machine learning are introduced. The possibility for creating some libraries with open machine learning models is also discussed.

## 2. Design Process of Analog Devices

A similar approach to the Gajski–Kuhn Y-chart [22], which explains the characteristics of a design process through its three domains, outlining behavioral, structural, and physical design, is applied here (Figure 1a). The Gajski–Kuhn Y-chart is created with examples for digital circuit design, but in this work is adapted to the specificity of analog circuit design using machine learning (Figure 1b—X-chart).

- The behavioral domain in the Gajski–Kuhn Y-chart presents the function of a given circuit without knowing the components that are included for its implementation. In this domain, the electronic circuit is seen as a "black box", in which only its inputs and outputs are known.
- The structural domain defines how the circuit is built. It considers the circuit structure, building components and the connections between them. The structural domain provides one of the possible transformations of the behavioral description into a set of components and relationships between them, which satisfies the predefined user specification.
- The physical domain shows exactly how the circuit has to be implemented on the board layout in order to ensure the desired behavior of the circuit. The main problems here concern the component placement on the PCB and their routing, taking into account the constraints of the limited chip area, the specific features of the components

and their physical geometry, the routing collisions, and congestion. Physical design is a complex task and is currently performed in several steps: macro placement, global placement, detailed placement, global routing, and detailed routing.

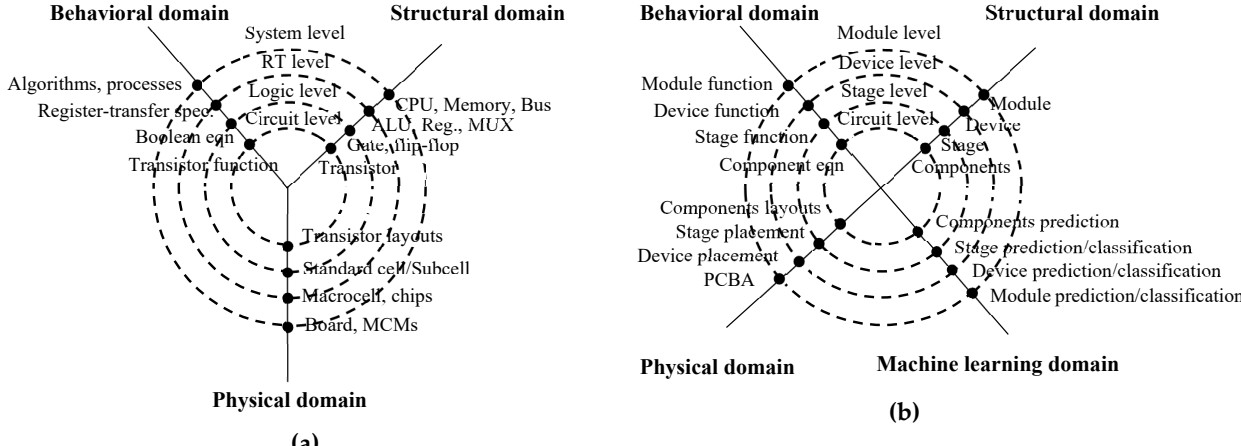

**Figure 1.** Design process depicted by: (**a**) Gajski-Kuhn Y-chart [22]; (**b**) proposed X-chart for analog design through machine learning.

In the proposed X-chart, the design process is supported through usage of machine learning as a new domain of machine learning is added. At the behavioral domain in the X-chart, the stage, device, or module are examined as a "black box" and the designer is interested only in their input and output, but not in the circuit structure. The structural domain explains the exact circuit topology of stages, devices, and modules as a given stage is created through components (transistors, resistors, capacitors, diodes, etc.). One device can be built through one or several stages and a module includes more than one device. The physical domain presents the best placement of the components, stages, devices, and modules on the PCB and their routing, forming printed circuit board assembly (PCBA). The machine learning domain shows the supportive role of machine learning algorithms in the design process, assisting the designers to accurately solve the specified problem and to make the correct decisions. This domain reveals the capability of machine learning in the prediction/classification of suitable components for realization of a stage, prediction/classification of possible stages for an electronic circuit design of a device, and prediction/classification of the possible devices on the PCB that form a module. For a given behavior of the circuit, several structural and physical implementations are possible and machine learning is applied to find possible solutions and the best approach. For this purpose, libraries with open machine learning models of circuits are prepared and used to shorten the design process and increase the design quality.

The chosen design strategy in this research is from bottom to top with hierarchical dependence between its four phases. The framework for circuit design through machine learning is shown in Figure 2:

(1) The first phase identifies suitable components for analog circuit creation. The common added components are transistors, resistors, capacitors, diodes, etc., which are organized in libraries. The electrical behavior of components is described with equations. The created machine learning (ML) models, which are also organized in libraries, are capable of predicting and classifying possible components for circuit implementation of a given stage.

(2) The second phase determines the appropriate stages that can form the circuit device. In the case of amplifier design, the circuit can be built from one stage, which is called a single-stage amplifier; a circuit built from two stages is known as a two-stage amplifier; and a circuit built from more stages is known as a multi-stage amplifier. An amplifier stage includes an amplifier element (here are considered just transistors), a circuit

for connecting to the signal source, a power supply circuit, a circuit for ensuring the constant current mode, and a circuit for connecting to the load. It may also contain a circuit for implementing feedback in order to improve or change the parameters and characteristics of the stage. The schematics of all stages are organized in libraries. Machine learning is used for predicting/classifying the behavior, and the structure of possible stages through equations and transfer functions, as machine learning models are placed in a library.

(3) The third phase connects the identified stages forming a device. Some additional circuits may be added as common feedback or circuits for correction. The most commonly used devices form device libraries. Machine learning models predict/classify the behavior and structure of the device, in addition to its placement and routing on the PCB, taking into account the device function.

(4) The fourth phase demonstrates the realization of more complex electronic products, i.e., so-called modules. One module may consist of one or several devices, which are connected to realize the predefined user specification. Some additional circuits for parameter improvement and correction can be added. Machine learning predicts/classifies the behavior and structure of modules and device placement on the PCB, considering the devices' transfer functions and the function of the whole module.

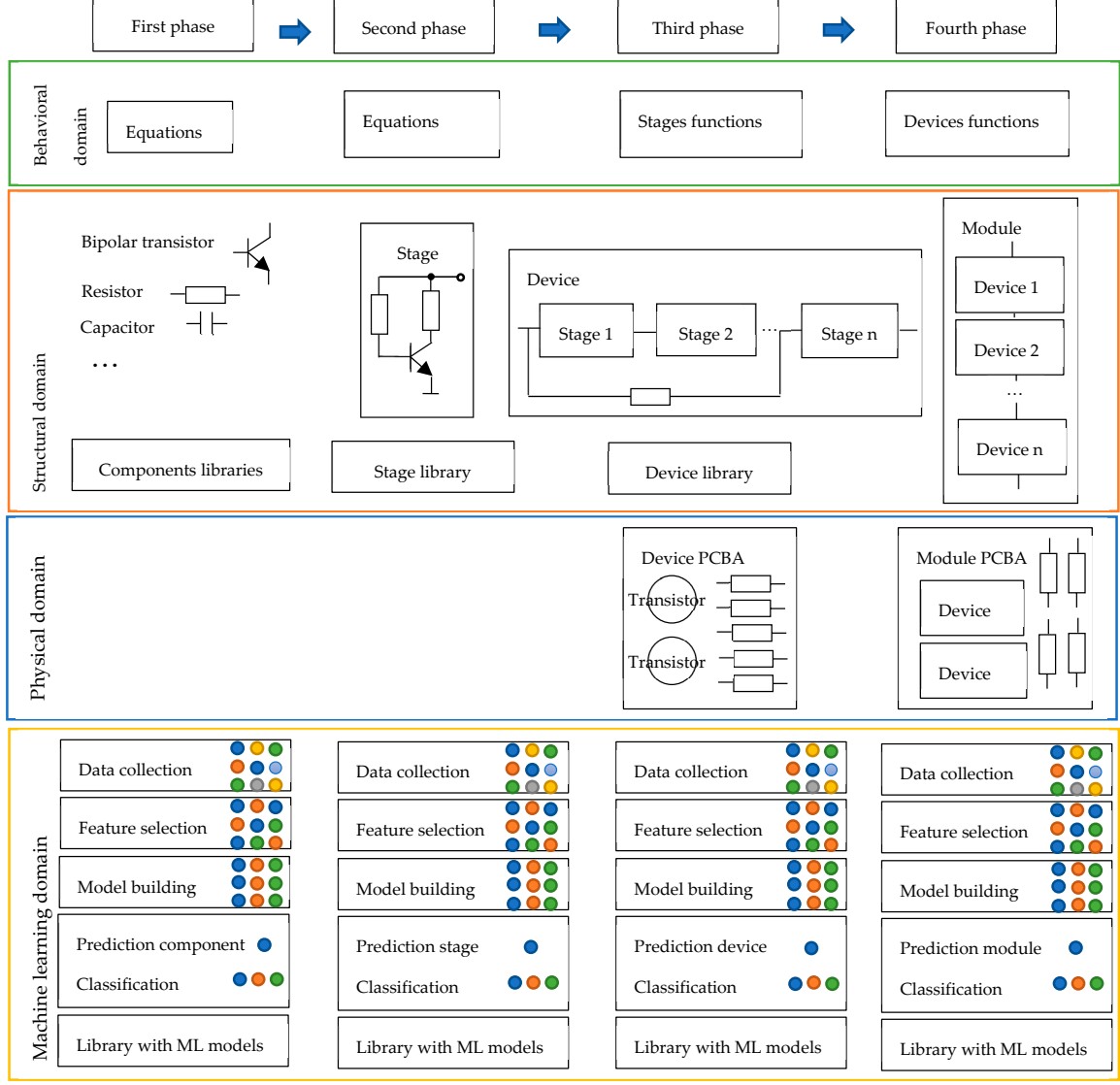

**Figure 2.** Framework for analog circuit design using machine learning.

### 3. Design of Analog Amplifiers

The accent in this work is on the third phase in the design process and on behavioral and structural modeling of analog amplifiers through machine learning. The components' prediction/classification and PCBA design will be within the scope of future work. The design of amplifiers can be carried out considering the specifics of a certain technology [23–25]. The most often used technologies are: bipolar; Bipolar-Field Effect Transistor (BiFET), where bipolar and MOS elements are formed in a common substrate; and Complementary Metal Oxide Semiconductor (CMOS), which uses complementary pairs of transistors. Each technology has its advantages and disadvantages, and is preferred for applications in specific cases. Bipolar technology is characterized with the possibility to obtain large voltage gain, small unbalanced input voltage, and very low noise voltage. In BiFET amplifiers, the input transistors are FETs and the rest of the circuit is made up of bipolar transistors. These are characterized by a higher rate of rise in the output voltage compared to bipolar and CMOS amplifiers. The common characteristic of bipolar and BiFET technology is that they allow for a wider bandwidth compared to CMOS technology. The advantages of CMOS amplifiers are related to operation at lower supply voltages, using mainly one supply voltage, and providing an operating range of the input and/or output voltage that is approximately equal to the supply voltage (rail-to-rail mode), as the consumed current is kept at a small value. It is obvious that the design process of amplifiers depends on the features of the technology. In this paper, the design of bipolar amplifiers is examined, and the CMOS design methodologies, because of their contemporary interest, will be discussed and explored in future work.

At the beginning, a library with the most common amplifier stages is introduced with their transfer functions and some parameters. According to a definition, an amplifier is an electronic device used to amplify an electrical signal in terms of current, voltage, or power. It is a converter of the electrical energy of the voltage supply source $V_{CC}$ into another type of electrical energy suitable for delivering to the load in its output circuit. In amplifier circuits, the signal transmission is carried out from the input to the output, but it is possible to use one or several feedback circuits. The most important parameters of amplifiers are the amplification coefficients, and input and output resistance. In multi-stage amplifiers, the overall transfer function has to be found taking into account the functions of the building stages, which can be categorized as input, intermediate, and output. Figure 3 presents a block diagram of a multi-stage amplifier, for which a wide variety of stages can be involved in its design to satisfy the requirements. In most cases, the aim of the first input stage is to obtain high gain and good suppression of the common-mode signals so that unwanted interference is not amplified and propagated to subsequent stages. Another requirement for the first stage is to provide a high input impedance. There are different variants for realization of this input stage, but in many cases a differential pair with or without a current source or through a cascode common emitter (CE)–common base (CB) is used. The role of the intermediate stage is to increase the amplifier gain, so it is very often realized through CE or a differential pair with or without a current source. The purpose of the driver stage is to provide appropriate values of currents and voltages to drive the output transistors. This requirement also predetermines its construction using a CE transistor or a Darlington transistor with or without a dynamic load. The output stage must provide a small output resistance and a certain output power. Therefore, it often involves a push–pull power amplifier circuit, with the (non-) complementary transistors being single and connected in a common collector (CC) circuit, or a Darlington transistor circuit can be used. In order to improve the parameters and characteristics of the amplifier, auxiliary circuits for realization of feedback and frequency domain correction can be introduced.

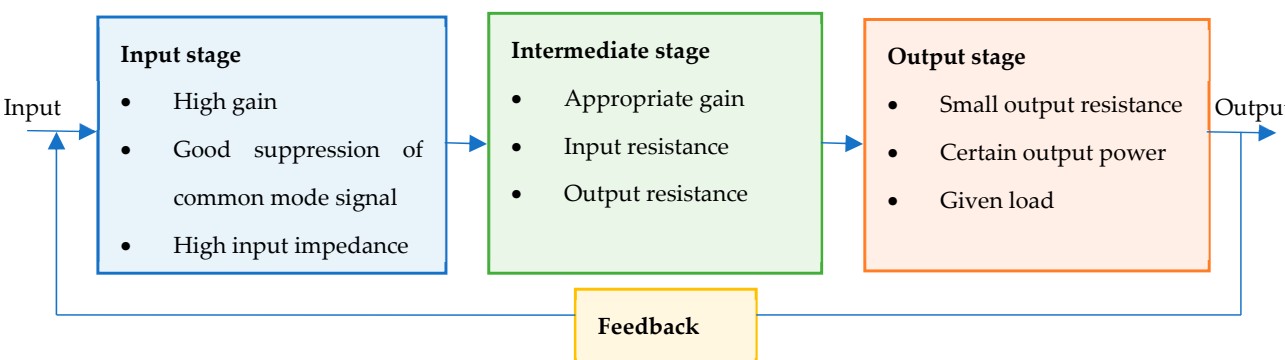

**Figure 3.** Structure of a multi-stage amplifier.

Let us suppose that the library with stages contains the most commonly used stages in amplifiers, some of which are presented on Figure 4. Their main parameters are summarized in Table 1 and they are: $A_V$—voltage amplification; $A_I$—current amplification; $A_d$—differential coefficient of voltage amplification; $r_{iA}$ and $r_{oA}$ are respectively input and output resistance of the stage; $R_C$ and $R_E$ are resistors in collector and emitter circuits, respectively; $r_{BE}$ and $r_{CE}$ are input and output resistance of a transistor, respectively; $g_m$ is transconductance; CMRR—Common Mode Rejection Ratio.

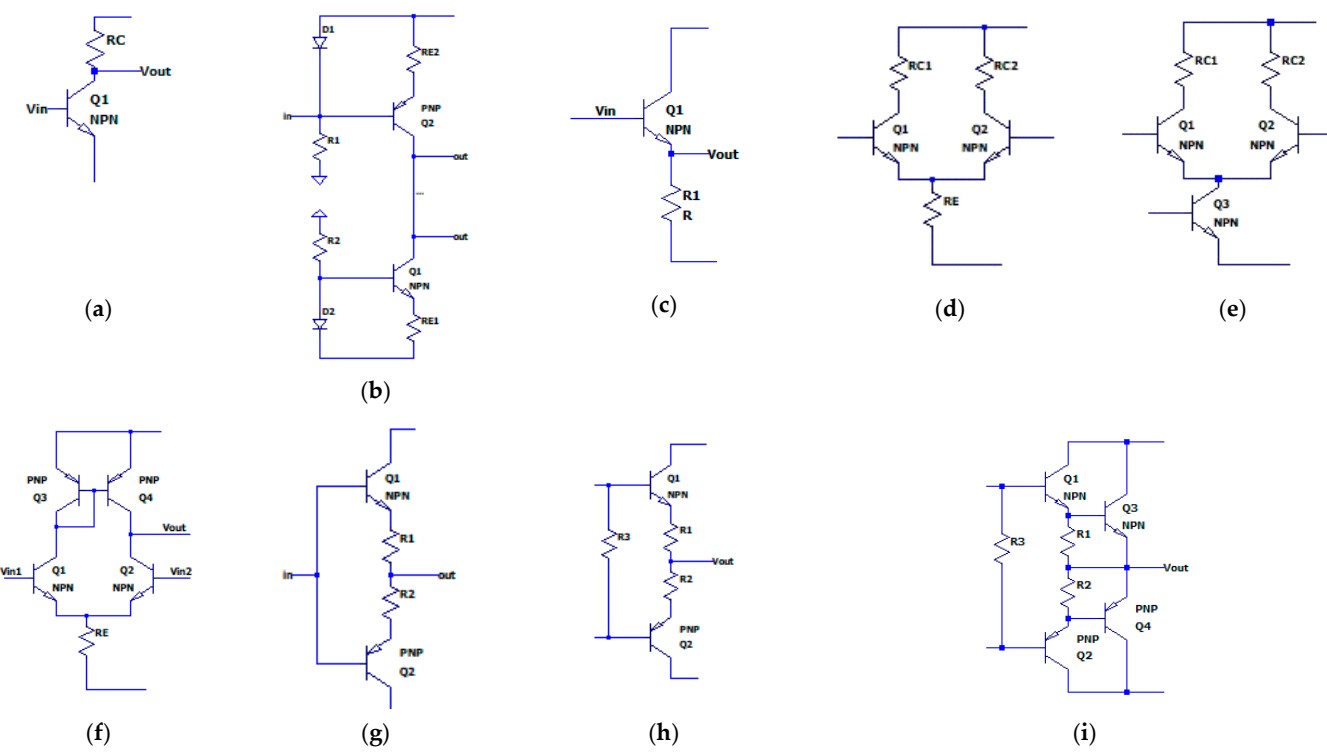

**Figure 4.** (**a**) Common emitter; (**b**) common emitter with active load; (**c**) common collector; (**d**) differential pair with resistive load and emitter resistor; (**e**) differential pair with resistive load and current source; (**f**) differential pair with active load; (**g**) complementary output pair in class B; (**h**) complementary output pair in class AB; (**i**) complementary output pair with Darlington transistors in class AB.

**Table 1.** Parameters of amplifier stages.

| Stage | Parameters | Function | Type |
|---|---|---|---|
| (a) Stage 1: common emitter | $A_V = -g_m(r_{CE}||R_C||R_L) = -g_m R'_{oA}$ ⁻high $A_I = \frac{i_L}{i_i} = -\frac{r_i}{R'_{oA}}A_V$ ⁻high $r_{iA} = R_B||r_i \approx r_{BE}$ ⁻medium $r_{oA} = r_o||R_C \approx R_C$ —medium | Amplifies voltage, current, and power, inverts the phase of the input voltage by 180° | Intermediate |
| (b) Stage 2: common emitter with active load | $A_V = -g_m(r_{CE_1}||r_{CE_2})$ ⁻higher $A_I$ ⁻high $r_{iA} = r_{BE_1}$ ⁻medium $r_{oA} = r_{CE_1}||r_{CE_2}$ —high | Amplifies voltage, current, and power, possesses increased amplification gains | Intermediate |
| (c) Stage 3: common collector | $A_V \approx \frac{g_m R_E}{1+g_m R_E} < 1$ ⁻does not amplify $A_I \approx h_{21e}$ ⁻high $r_{iA} \approx R_B||h_{21e}R_E$ ⁻high $r_{oA} = \left(\frac{1}{g_m} + \frac{R_G}{h_{21e}}\right)||R_E$ —low | Repeats the input voltage (voltage follower), but amplifies the current and power | Output |
| (d) Stage 4: differential pair with resistive load and emitter resistor | $A_d \approx g_m R_C||\frac{R_L}{2}$ ⁻high $r_{id} \approx 2r_{be}$ ⁻high $r_{od} \approx 2R_C$ ⁻medium $CMRR \approx 2g_m R_E$ —high | Amplifies the difference between both inputs | Input |
| (e) Stage 5: differential pair with resistive load and current source; | $CMRR = \frac{2g_{m_3}(1+g_{m_3}R_3)}{h_{22}}$ —higher | Better suppression of common mode signals | Input |
| (f) Stage 6: differential pair with active load | $A_d \approx (r_{02}||r_{04}||R_L)$ —higher | Higher differential gain is achieved through adding active load | Input |
| (g) Stage 7: push-pull stage with complementary output pair in class B | $A_V < 1$ ⁻does not amplify $P_L = \frac{1}{2}U_L I_L = \frac{1}{2}I_L^2 R_L$ $\eta = \frac{P_L}{P_{CC}} = \frac{\pi}{4}\frac{U_L}{U_{CC}}$ $\eta_{max} = \frac{\pi}{4}\frac{U_{Lmax}}{U_{CC}} = \frac{\pi}{4} \approx 0.785$ or $78.5\%$ | Each of the transistors operates in an CC circuit, which achieves high input and low output resistance, high current gain and low distortion. | Output |
| (h) Stage 8: complementary output pair in class AB | $P_{omax} = \frac{2U_{CC}-U_{BE}}{R_{1,2}}$ $\eta \approx 40$–$50\%$ | The resistor R3 is used for creating a bias voltage on the bases of transistors T1 and T2 | Output |
| (i) Stage 9: complementary output pair with Darlington transistors in class AB | $\eta \approx 40$–$50\%$ | The two diodes, in addition to creating a bias voltage on the bases of transistors T1 and T2, are also used to stabilize their operating current | Output |

## 4. Proposed Method

Behavioral design of amplifiers sees stages as "black boxes" and is not interested in exactly how they are implemented. It is important only to know the input and output parameters. Structural design is related to explanation of the possible structure and this task is multi-variant. Following the configuration from Figure 3, the amplifier has to possess input, intermediate, and output stages. It is known that the output stage does not amplify the voltage signal, but it is responsible for small output resistance. This means that the input and intermediate stages have to deliver the required amplification. The feedback configuration is considered to be the same in the amplifier design.

Machine learning algorithms as a part of artificial intelligence have recently been utilized to assist in the engineering tasks related to the design process in electronics, integrating some approaches in EDA software. Huang et al. explored the contemporary scientific achievements in this area and reported enormous interest in automating a wide variety of engineering activities through the usage of machine learning in different EDA tools [26]. Ren noted the role of machine learning in solving multiple EDA problems [27].

He summarized the applications of various machine learning algorithms for more efficient workability of EDA software and for improving designer efficacy.

In this work, a machine learning-driven approach was used to study data about the stage type—input, intermediate, and output—assisting the designer to make a choice about the type of the most suitable stages for realization of a three-stage amplifier. Moreover, machine learning models predict some important parameters of each stage type. The proposed method for amplifier design through machine learning is presented on Figure 5. It is a two-step predictive method: in the first step, the stage type is predicted, and in the second step, some typical parameters for each stage type are forecasted. The suggestion is that a library with amplifier stages exists and data regarding the function and structure of each stage are gathered. Datasets are learnt by supervised machine learning algorithms, including those for rules extraction, which results in models capable of predicting the stage type (input, intermediate, output) and some main parameters of each stage.

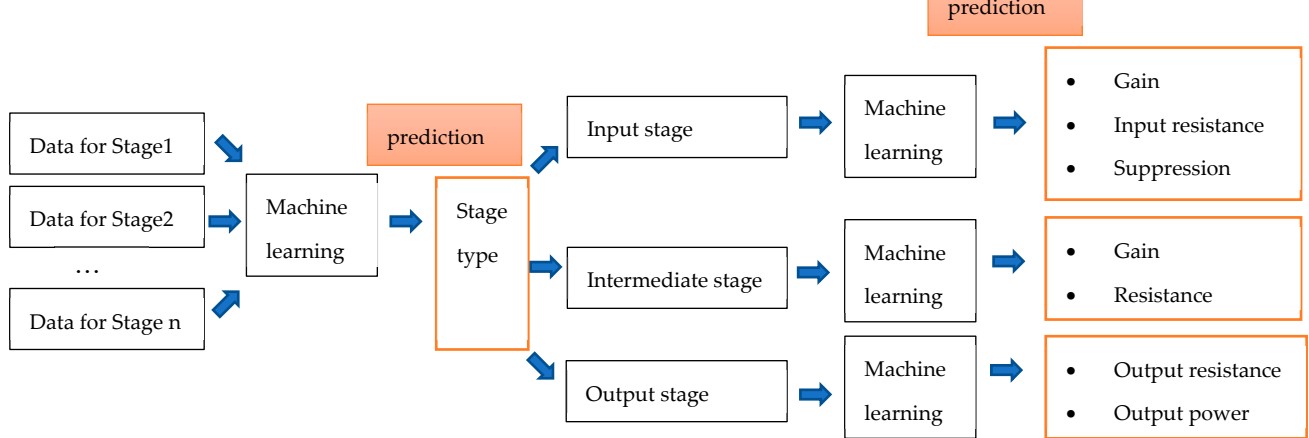

**Figure 5.** Method for amplifier design through machine learning.

*Datasets Preparation*

The dataset, presented in Table 2, is formed considering 30 different stages and their main parameters. It is used in the first step from the proposed method to predict the stage type and stage logic.

**Table 2.** Dataset for prediction of the stage type.

| $A_V$ | $A_I$ | $r_{id}$ | $r_{od}$ | CMRR | $P_L$ | Circuit | Stage Type |
|---|---|---|---|---|---|---|---|
| high | n/a | high | medium | high | n/a | diff. pair with resistive load | input |
| higher | n/a | high | medium | high | n/a | diff. pair with active load | input |
| high | n/a | high | medium | higher | n/a | diff. pair with current source | input |
| high | high | medium | medium | n/a | n/a | common emitter | intermediate |
| higher | high | medium | high | n/a | n/a | common emitter with active load | intermediate |
| does not amplify | high | high | Low | n/a | medium | push-pull class AB | output |
| … | … | … | … | … | … | … | … |

For realization of the second step of the method, datasets of the parameters included in the library stages are prepared. All values of parameters are received after mathematical calculations and certain methodologies for analog design with bipolar transistors, as follows [28–31]. The used methodologies are summarized through different algorithms. For example, Algorithm 1 shows the main calculations for obtaining the parameters of output stage 8. Algorithm 2 is applied for collecting the data for intermediate stage 2 and Algorithm 3 for the input stage 4. Respectively, Tables 3–5 present a part of the gathered

datasets for output stage 8 (with 161 records), intermediate stage 2 (with 988 records), and input stage 4 (451 records) according to defined Algorithms 1–3. The datasets for other stages are gathered in a similar way. For demonstration, the power transistors 2SCR587D3 and 2SAR586D3 [32,33] (for output stage), middle power transistors 2SCR563F3 and 2SAR563F3 [34,35] (for intermediate stage), low power transistor 2N3904 [36] (for input stage), and diode 1N5819 [37] are chosen.

**Table 3.** Parameters of output stage.

| $V_{R1}$, V | $V_{R2}$, V | $I_{Lm}$, A | $R_1$, $\Omega$ | $R_2$, $\Omega$ | $R_3$, $\Omega$ |
|---|---|---|---|---|---|
| 1.2 | 1.2 | 1.58 | 0.75 | 0.75 | 24 |
| 1.1 | 1.1 | 1.58 | 0.69 | 0.69 | 23 |
| 1 | 1 | 1.58 | 0.63 | 0.63 | 22 |
| ... | ... | ... | ... | ... | ... |

**Table 4.** Parameters of intermediate stage.

| $I_{E1}$, mA | $I_{E2}$, mA | $R_{E1}$, k$\Omega$ | $R_{E2}$, k$\Omega$ | $R_1$, k$\Omega$ | $R_2$, k$\Omega$ |
|---|---|---|---|---|---|
| 14.7 | 14.7 | 1.63 | 1.63 | 42 | 42 |
| 15.19 | 15.19 | 1.57 | 1.57 | 42 | 42 |
| 15.68 | 15.68 | 1.53 | 1.53 | 42 | 42 |
| ... | ... | ... | ... | ... | ... |

**Table 5.** Parameters of differential pair.

| $R_E$, k$\Omega$ | $I_E$, mA | $R_C$, k$\Omega$ | $g_m$, mS | $A_d$ | $r_{id}$, k$\Omega$ | $r_{od}$, k$\Omega$ |
|---|---|---|---|---|---|---|
| 22.6 | 0.5 | 5.4 | 19.23 | 67.432 | 12 | 10.8 |
| 18.833 | 0.6 | 4.5 | 23.076 | 71.618 | 13 | 9 |
| 16.142 | 0.7 | 3.857 | 26.923 | 74.941 | 14 | 7.714 |
| 14.125 | 0.8 | 3.375 | 30.769 | 77.641 | 15 | 6.75 |
| 12.555 | 0.9 | 3 | 34.615 | 79.881 | 16 | 6 |
| 11.3 | 1 | 2.7 | 38.461 | 81.768 | 18 | 5.4 |
| ... | ... | ... | ... | ... | ... | ... |

---

**Algorithm 1:** Design of output stage
Preliminary data: load resistance $R_L = 8\,\Omega$, output power $P_L = 10$ W, voltage supply $V_{CC} = 12$ V; choice of power transistors and their parameters, taken from datasheet specifications [32,33]

1. Calculating the voltage on the load $V_{Lm} = \sqrt{2P_L R_L}$;
2. Calculating the current through the load $I_{Lm} = \frac{2P_L}{V_{Lm}}$;
3. Calculating $R_{1,2} = \frac{V_{R1,2}}{I_{Lm}}$ $(V_{R1,2} \leq 0.1 V_{Lm})$;
4. Calculating $R_3 = \frac{V_{R3}}{I_{C,Q} - I_{C,Q}/h_{21,Q}}$, $\left(I_{C,Q} = (0.01 \div 0.05)I_{Lm}\right)$.

---

**Algorithm 2:** Design of intermediate stage
Preliminary data: taken from datasheet specifications [34,35]

1. Calculating $I_{E1,2} \approx (3 \div 5)\frac{I_{Lm}}{h_{21,1,2}}$;
2. Calculating $R_{E1,2} = \frac{V_{RE1,2}}{I_{E1,2}}$, $V_{RE1,2} \approx 2U_{CC}$;
3. Calculating $V_{D1,2} = V_{BE1,2} + V_{RE1,2}$;
4. Calculating $R_{1,2} = \frac{V_{CC} - V_{D1,2}}{I_{D1,2}}$.

---

**Algorithm 3:** Design of input stage

Preliminary data: choice of low power transistor and its parameters, taken from datasheet specification: $I_C$, $V_{CE}$, $h_{11}$, $h_{21}$, $h_{22}$, $g_m$ [36]

---

1. Calculating $R_E = \frac{V_E - (-V_{CC})}{I_E}$, $(V_{E1} = V_{E2} = V_E = -V_{BE}$ and $I_{E1} = I_{E2} = I_E \approx I_C)$;
2. Calculating $R_{C1} = R_{C2} = R_C = \frac{V_{CC} - V_C}{I_C}$, $(V_{C1} = V_{C2} = V_C = V_E + V_{CE})$;
3. Calculating $A_d = g_m R_C \left|\right| \frac{R_L}{2}$;
4. Calculating $r_{id} \approx 2r_{be}$;
5. Calculating $r_{od} \approx 2R_C$;
6. Calculating $CMRR \approx 2g_m R_E$.

---

## 5. Results

To verify the proposed two-step method and applicability of machine learning in support of amplifier design, the functions and structure of the stages and the collected data, as presented in Tables 1–5 and the schematics in Figure 4, are considered. Let us suppose that the designer is required to build an amplifier with the following parameters: input resistance $r_{iA} = 7$ kΩ, load resistance $R_L = 8$ Ω, output power $P_L = 10$ W, amplification $A = 1200$, voltage supply $V_{CC} = 12$ V. In the first step, the correct stages have to be chosen to satisfy the formulated user requirements. For this purpose, the designer can rely on machine learning predictions regarding which stage is suitable for usage as an input, intermediate, and output stage. Moreover, logic generated by rule induction algorithms can support their decision making.

For the dataset from Table 2, the Decision Tree algorithm was applied in the environment of RapidMiner Studio [38]. The created model for prediction of the stage type is characterized with 89.74% accuracy for the ratio of training/testing data of 60%/40%.

Figure 6 presents the probability of correct predictions, which is given through confidence (confidence for predicting input, intermediate, and output stages). A larger value of confidence (the maximum value is 1 and the minimum value is 0) means a greater probability of true correct predictions. It can be seen that the confidence of input stages is 1, while the confidence of intermediate and output stages is smaller than 1.

For the same dataset, two algorithms for rule extraction were applied: Rule induction and Trees to rules. Through Rule induction machine learning techniques, several formal rules can be generated in the form if–then–else, driven by the collected data. The advantages of these techniques lead to a better explanation and understanding the logic of the examined problem [39,40], in our case, the amplifier construction. When the first algorithm rule induction is applied, the following result is obtained:

---

If $R_{out} = low$, then it is an Output stage;
If $R_{in} = high$, then it is an Input stage;
else it is an Intermediate stage.

---

These extracted rules indicate the stage type according to some typical parameters. The exploration of the generated logic says that, if the output resistance of a given stage is low, then this stage is suitable as an output stage; if the input resistance is high, then this is an input stage, and, in other cases, the stage is intermediate.

At the application of the second algorithm, Trees to rules, the achieved results outline another rule logic for identification of the stage type. If CMRR is high, then this is the input stage. If CMRR is not an important parameter for a given stage and input resistance is high, then the stage is an output stage. If the input resistance has a medium value and CMRR is not an important parameter, then the stage is intermediate.

---

If $CMRR = high$, then it is an Input stage;
If $CMRR$ is not defined and $R_{in} = high$, then it is an Output stage;
If $CMRR$ is not defined and $R_{in} = medium$, then it is an Intermediate stage.

---

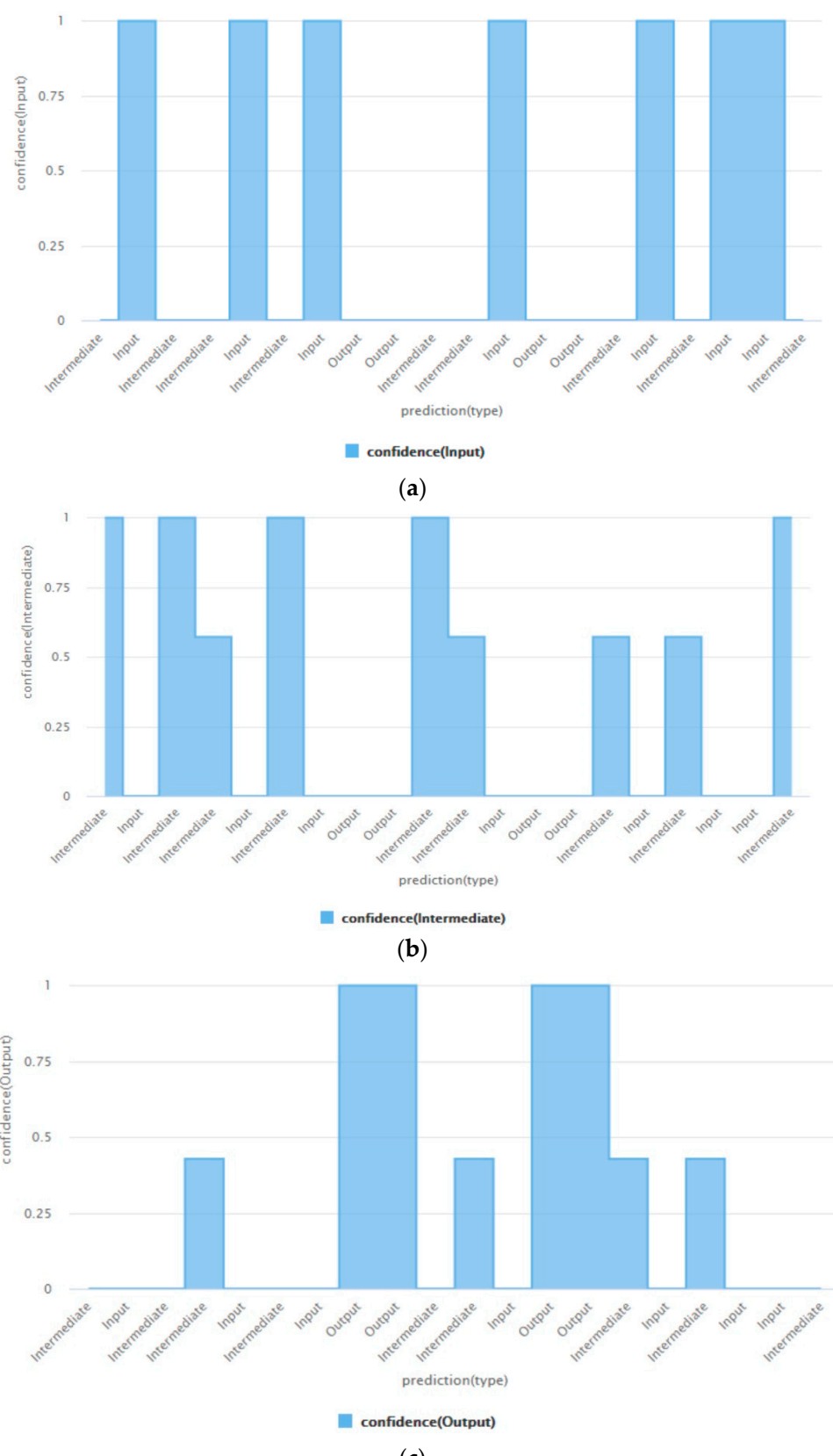

**Figure 6.** *Cont.*

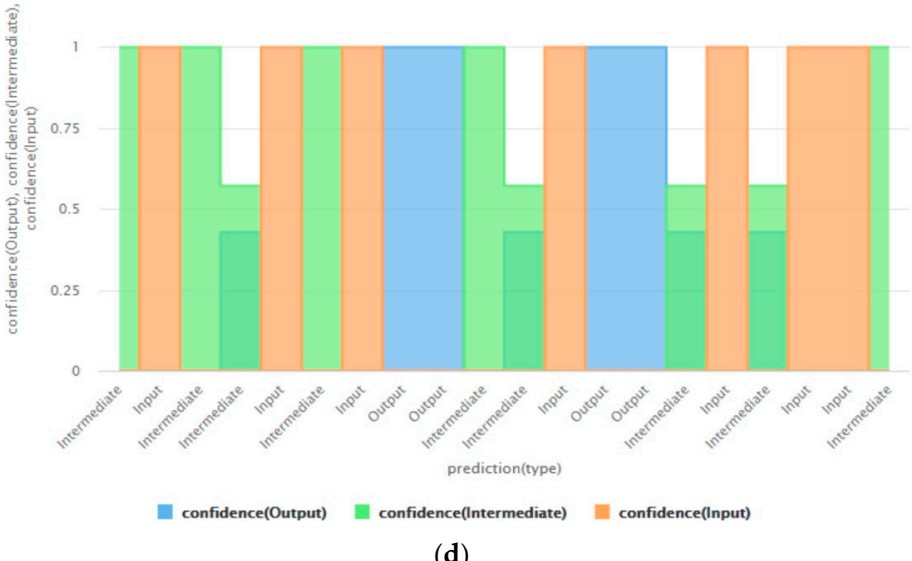

(**d**)

**Figure 6.** Confidence of predicted stage type: (**a**) input stage; (**b**) intermediate stage; (**c**) output stage; (**d**) comparison of confidence for different stage types.

The findings indicate that the algorithms for rules induction are very useful for data mining and knowledge discovery in the area of electronics, as the generated logic can be a supportive tool and easily integrated in EDA software. The automatic generation of formal rules and formalization of the process of analog circuit design can be considered to be an advantage for designers.

In the second step, four machine learning algorithms are used: Decision Tree, Random Forest, Gradient Boosted Tree, and Support Vector Machine [41,42], to find the best model for prediction of the parameters of different types of stages. Machine learning models are created taking into account the datasets for each stage considering its typical parameters.

Figure 7 presents only the prediction charts of the created predictive models for input stage 4. It can be seen that the best solution for this regression task is the Support Vector Machine algorithm. Similar results are obtained for other stages.

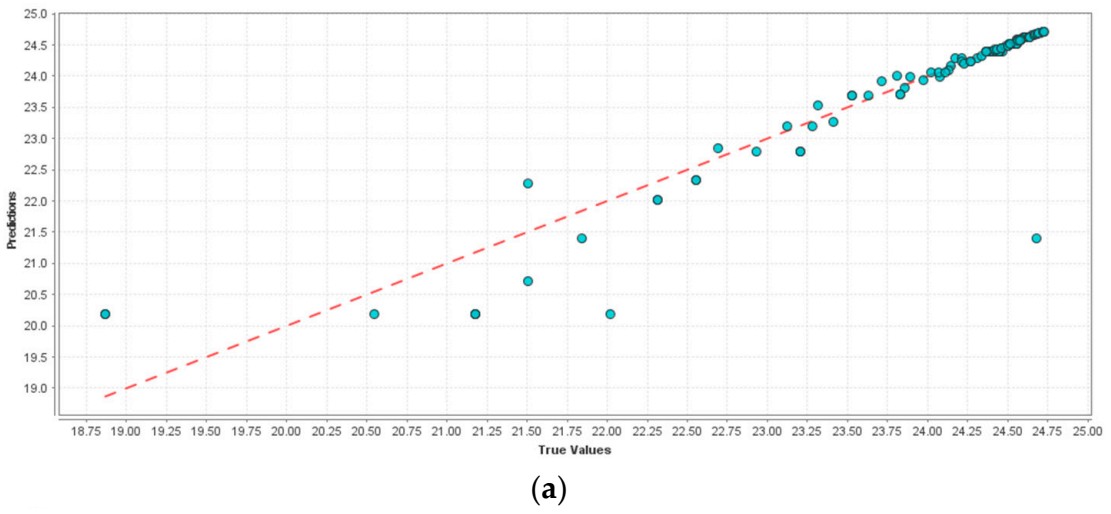

(**a**)

**Figure 7.** *Cont.*

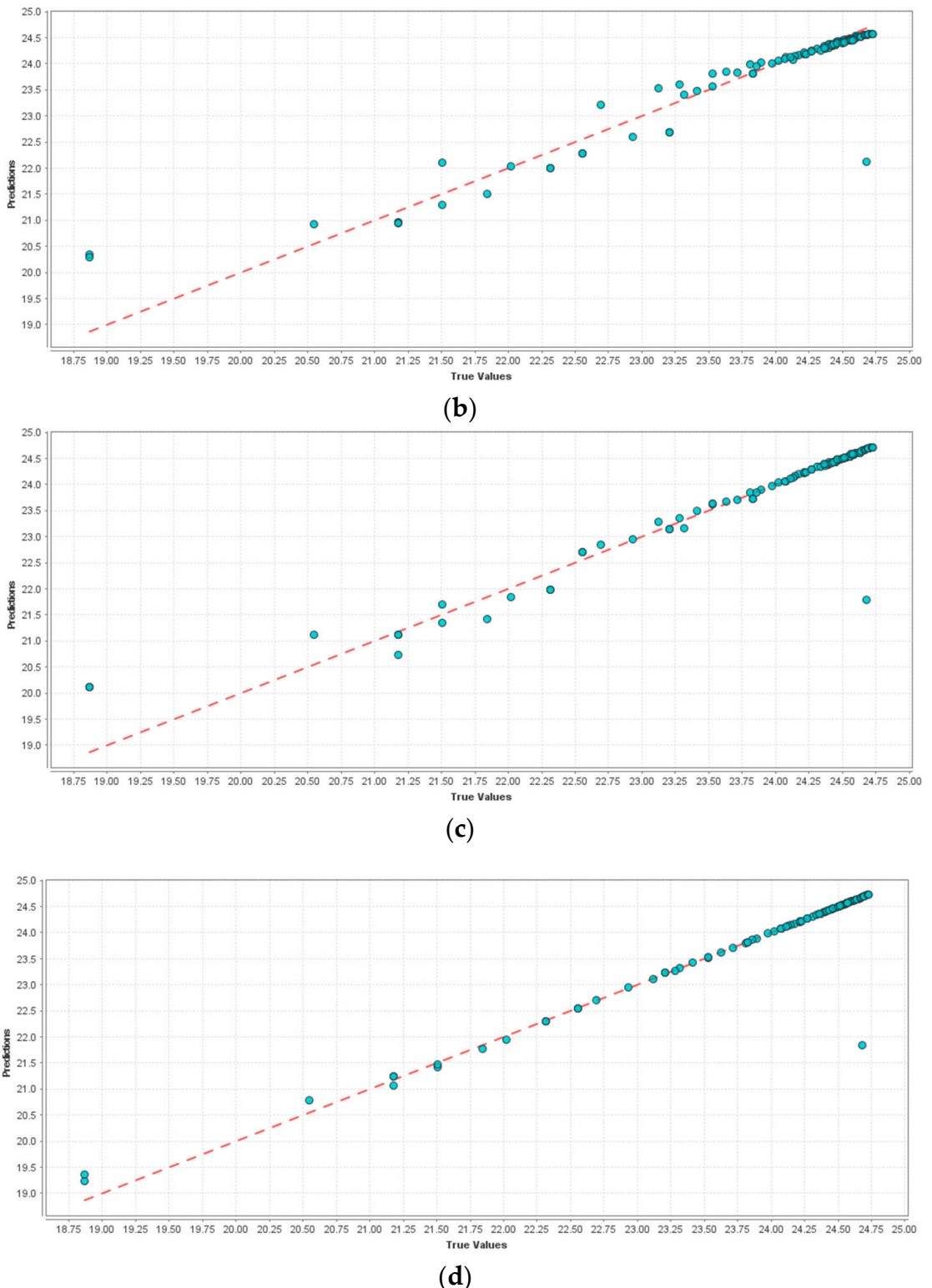

**Figure 7.** Prediction charts: (**a**) Decision Tree; (**b**) Random Forest; (**c**) Gradient Boosted Trees; (**d**) Support Vector Machine.

The learners are compared as they were evaluated using standard metrics for machine learning through parameters: root mean square error (RMSE), absolute error (AE), and squared error (SE) (Figure 8). The smallest errors were obtained for Support Vector Machine.

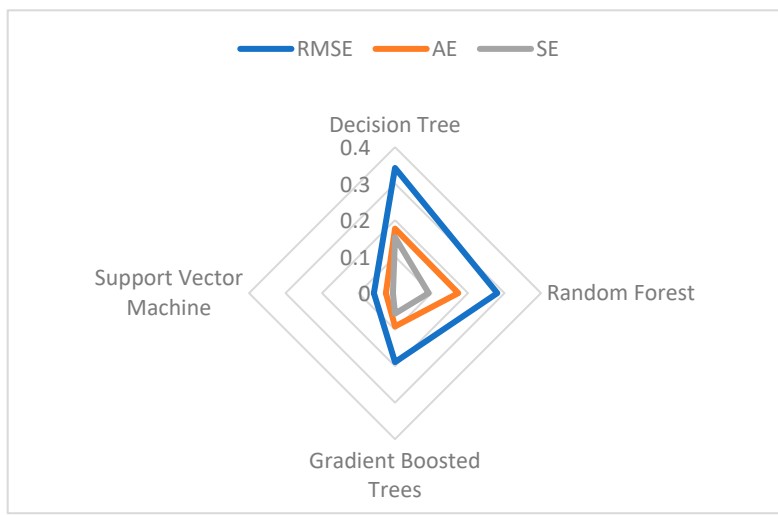

**Figure 8.** Comparison of the applied learners.

## 6. Case Study

This section demonstrates the amplifier building, taking into account the support received by machine learning and the results obtained in previous sections. The amplifier design begins at the back and moves forward, i.e., from the design of the output stage to the input and intermediate stages. The designer is supported in the first step with charts similar to that presented on Figure 9. Here, the designer can gather information about suitable stages for usage as output stages. It can be seen that these are: complementary output pair in class B, complementary output pair in class AB, and complementary output pair with Darlington transistors in class AB. Then, the designer will decide to use the complementary output pair in class AB, because of the obtained information from the rules logic created by applying rule induction machine learning algorithms. This decision is also in line with the user's predefined specification, in which there are no additionally defined parameters for the stage, apart from the output power and the load. This gives the designer the possibility of choosing a simpler stage with a smaller number of components that is capable of satisfying the user requirements.

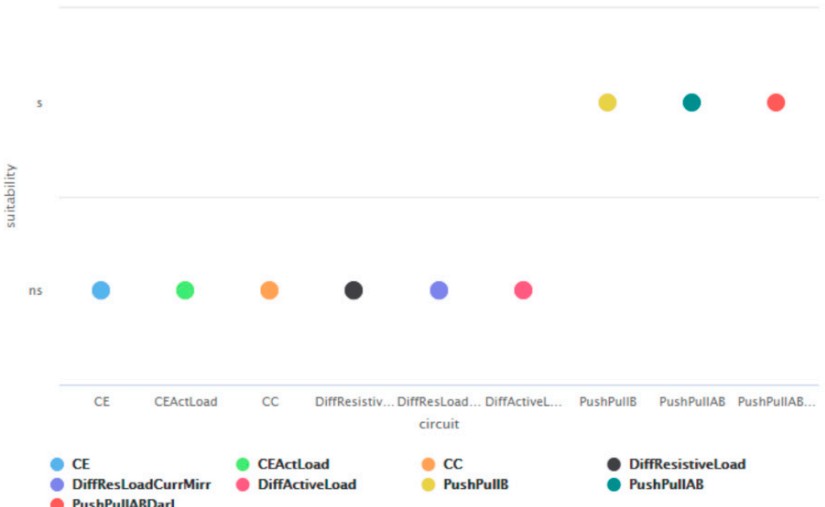

**Figure 9.** Dependence between circuits of stages and their suitability for realizing the amplifier output (s—suitable, ns—not suitable).

In the second step, for the output stage, $P_L$ and $R_L$ are known and the designer has to obtain the values of the included resistors $R_5$, $R_6$, and $R_7$, which can easily be taken from the prediction chart for the output stage presented in Figure 9. There is no need for the designer to perform the calculations presented in previous sections or to recreate the datasets for the stages in the library. Once the machine learning models are created, they can be used repeatedly. The designer only needs to use the machine learning results and the predicted values of the parameters. Let us suppose that the voltage value $V_{R5,6}$ is 1 V; then, the predicted resistor value $R_{5,6}$ of 0.63 $\Omega$ can be found from the prediction chart (Figure 10). Using a similar chart, the value of the resistor $R_7$ can be predicted, as here it is 22 $\Omega$ considering the operational regime of the transistor $T_3$. In practice some diodes can be used instead of $R_7$.

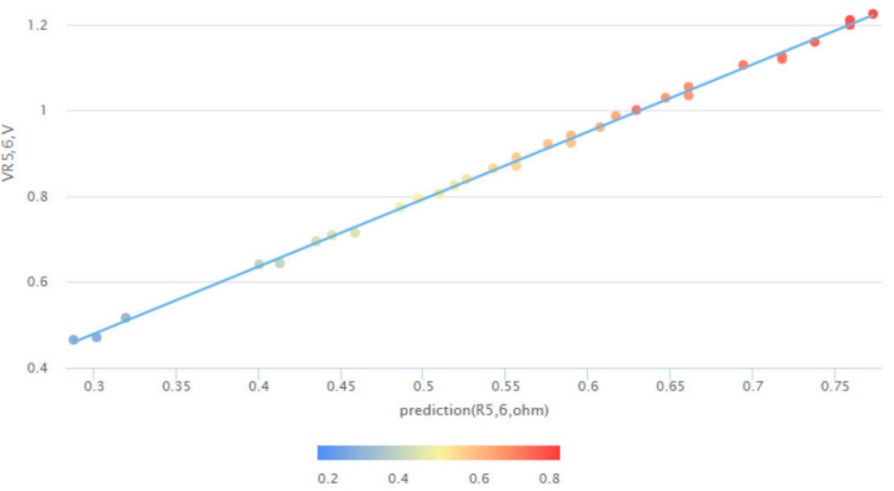

**Figure 10.** Predicted dependence between $R_{5,6}$ and $V_{R5,6}$.

The intermediate stage is designed taking into account Figure 11, where the designer can see the dependence between predicted resistors $R_{E3,4}$ at a given current $I_{E3,4}$. The resistors $R_{E3,4}$ are selected from the prediction chart to be 1.63 k$\Omega$ at the current of 14.7 mA. The resistors $R_{3,4}$ are chosen to have a value of 42 k$\Omega$ at a given current through the diodes $D_{1,2}$.

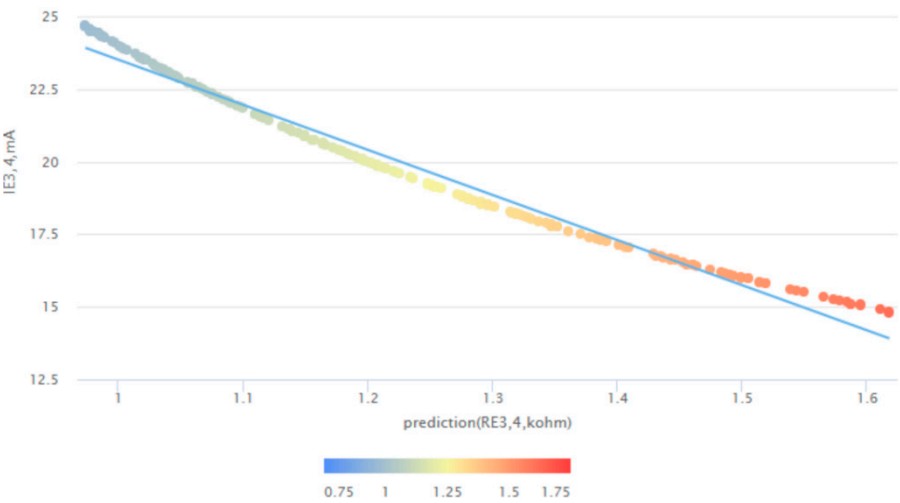

**Figure 11.** Predicted dependence between $R_{E3,4}$ and $I_{E3,4}$.

The design of the input stage is facilitated through the predicted chart on Figure 12, in which the dependence between $R_C$ and $R_E$ is presented. At the current of $I_E = 1$ mA, the predicted values of the collector and emitter resistors are respectively 2.7 and 11.3 kΩ.

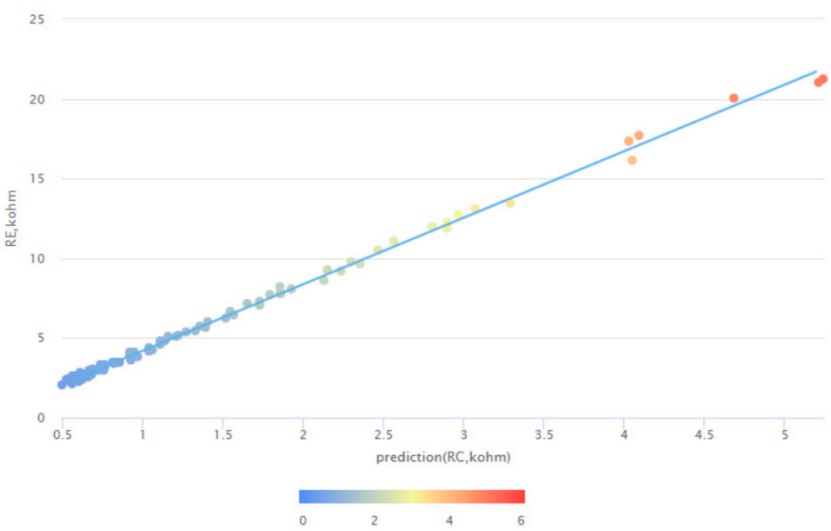

**Figure 12.** Predicted dependence between $R_C$ and $R_E$.

Finally, the negative feedback also has to be considered, because the user requires the amplification to be 1200. The feedback is realized through two resistors, as the resistance of the first $R_{F1}$ is chosen by the designer and the second $R_{F2}$ is calculated according to the equation: $A_{VF} = 1 + \frac{R_{F1}}{R_{F2}}$. The resistors $R_{F1} = R_{in}$ are selected with an appropriate value of 18 kΩ to match the input impedance of the amplifier. Then, $R_{F2}$ is calculated from the above-mentioned equation to be 15 Ω. The constructed amplifier according to the initial user's requirements and designer's choices, which are supported through machine learning, is presented in Figure 13.

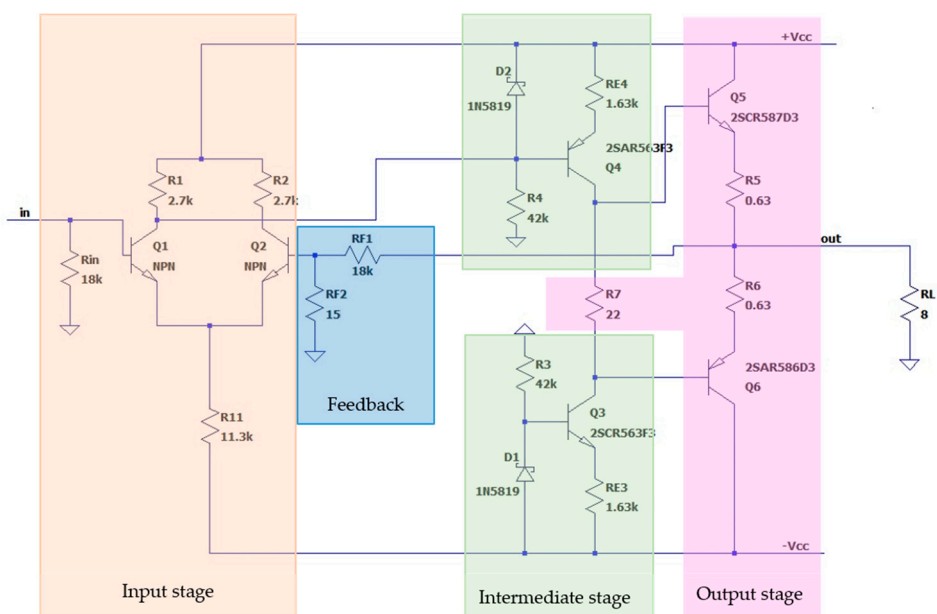

**Figure 13.** The constructed amplifier.

## 7. Conclusions

The paper presents a two-step method for facilitating the design of analog amplifiers using machine learning algorithms and rule induction techniques. In the first step, the designer is assisted with suggestions about the most suitable stages for realization of amplifiers considering the predictions regarding the stage types and generated rules logic. In the second step, some parameters of a given stage type are indicated to support the designer's choice regarding the most relevant stage according to the predefined user specifications. The method was verified in the design of a three-stage amplifier, for which the functions and main parameters of the building stages are known. As a learner in the first step, the Decision Tree algorithm, from supervised machine learning, was applied to solve the classification task and achieved the best accuracy of 89.74%. Extracted logic is also demonstrated through usage of two different rule induction algorithms. In the second step, four machine learning algorithms are employed to learn data about different amplifier stages and to solve a regression task. The smallest errors were found with the use of Support Vector Machine.

The concept of a library of open machine learning models of circuits is introduced to assist the designer in the important, complex, and time- and effort-consuming activities that are typical for the design process of analog circuits, devices, and modules at structural, behavioral, and physical levels.

The Gajski–Kuhn Y-chart is extended to an X-chart, considering the increasing importance of machine learning in the design process of electronic circuits, and is adapted to the design of analog circuits. A framework for analog circuit design, taking into account the possibility of machine learning to support almost all design phases at behavioral, structural, and physical level, is proposed.

**Author Contributions:** Conceptualization, M.I. and M.A.S.; methodology, M.I.; validation, M.I. and M.A.S.; formal analysis, M.I.; investigation, M.I. and M.A.S.; resources, M.I.; writing—original draft preparation, M.I.; writing—review and editing, M.I. and M.A.S.; visualization, M.I. All authors have read and agreed to the published version of the manuscript.

**Funding:** This research is supported by the Bulgarian National Science Fund in the scope of the project "Exploration the application of statistics and machine learning in electronics" under contract number КП-06-Н42/1.

**Conflicts of Interest:** The authors declare no conflict of interest.

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
