# Peer review of "Machine Learning and Rules Induction in Support of Analog Amplifier Design"

_computation, doi:10.3390/computation10090145_

Round 1

Reviewer 1 Report

This paper presents a two-step approach for analog amplifier design where "classical" and machine learning algorithms are combined. The approach seems to be interesting, however a lot of details are missing to be able to  analyse deeply the proposed idea.

First, the authors did not give any details about their machine learning algorithms: what is the size of the dataset, what kind of simulations were used to create it, what is the architecture of their neural network, its size, the learning time ...

The authors also did not give enough details on the circuit part: what is the used technology, what are the targeted specifications, what features are used for learning, what are the parameters that are tweaked, ...

One of the specificity of this work is using bipolar transistors, i would have appreciated more details regarding this point and how does it compare with CMOS.

On the form:

The writing, in my opinion, needs to be improved. There are some typos, problems also prepositions, missing comas...

The quality of figures 5 and 6 needs to be improved. I needed to zoom at 300% to examine them. BTW, there are 2 figures 5

Author Response

Dear Reviewer, Thank you so much for your valuable comments and suggestions for the paper improvement! We are very grateful and thank you again!

This paper presents a two-step approach for analog amplifier design where "classical" and machine learning algorithms are combined. The approach seems to be interesting, however a lot of details are missing to be able to analyse deeply the proposed idea.

First, the authors did not give any details about their machine learning algorithms: what is the size of the dataset, what kind of simulations were used to create it, what is the architecture of their neural network, its size, the learning time ...

  • The size of the datasets is pointed out.;
  • The methodologies for analog design are summarized in the form of algorithms and some literature sources for analog circuit design are included. In this work the datasets are collected taking into account the mathematical descriptions of the stages.;
  • In this work, the neural networks are not used. For data study tree-based algorithms and support vector machine algorithm are applied.

The authors also did not give enough details on the circuit part: what is the used technology, what are the targeted specifications, what features are used for learning, what are the parameters that are tweaked, ...

  • A paragraph with the most used in practice technologies is added with their advantages and disadvantages.;
  • The specifications of used components are included in the literature list and they are cited in the paper to explain the demonstrative example. The main parameters and features for learning are also explained. The most important are placed in Tables.

One of the specificity of this work is using bipolar transistors, i would have appreciated more details regarding this point and how does it compare with CMOS.

  • A paragraph with the most used technologies for electronics manufacturing is added and our choice to use bipolar technology is argued. Also, our intention is in the near future CMOS technology to be investigated in the context of electronic circuit design with its specifics.

On the form:

The writing, in my opinion, needs to be improved. There are some typos, problems also prepositions, missing comas...

  • The paper is checked for grammar errors and typos.

The quality of figures 5 and 6 needs to be improved. I needed to zoom at 300% to examine them. BTW, there are 2 figures 5

  • The quality of all figures is improved and its numeration is checked.

Reviewer 2 Report

The manuscript provides a machine learning tool for the analysis and design of analogs amplifiers. The method is based on multi-stage approach by modifying the original X-chart of Gajski-Kuhn. In the opinion of this reviewer, the contribution could be a useful addition.

Yet, the manuscript is not recommended for the publication in the present form.

1-Substantial revisions are recommended regarding to the style and writing.  There are several  inconsistencies and errors that need attention. Proofreading is highly recommended. Also, please make sure there is no typos. For examples;

·     pp 1, line 31: ‘’and’’ generators…

·       pp 1, line 37: ‘’are’’ utilized…

·       pp 2, line 52: ‘’proposed’’ a compositional...

·       others

2-    as stated at the end of Section 2 (pp 3) the physical domain was not discussed. However, brief description could help the reader and thus is recommended.

3-    The accuracy of the classification process has been found to be enhanced by decreasing the portion of training data set (pp 7 lines 233-234. This finding may be contrary to the common practice. It is well known that the machine learning model is trained, and constructed based on the training set. That’s mean that the accuracy is expected to enhance as long as the percentage of the training data increases. Could the authors comment and justify their findings.

4- It is recommend to cite recent publication of machine learning in: Applied Sciences 11.18 (2021): 8762. DOI:https://doi.org/10.3390/app11188762

Author Response

Dear Reviewer, Thank you so much for your valuable comments and suggestions for the paper improvement! We are very grateful and thank you again!

The manuscript provides a machine learning tool for the analysis and design of analogs amplifiers. The method is based on multi-stage approach by modifying the original X-chart of Gajski-Kuhn. In the opinion of this reviewer, the contribution could be a useful addition.

Yet, the manuscript is not recommended for the publication in the present form.

1-Substantial revisions are recommended regarding to the style and writing.  There are several  inconsistencies and errors that need attention. Proofreading is highly recommended. Also, please make sure there is no typos. For examples;

  • pp 1, line 31: ‘’and’’ generators…
  • pp 1, line 37: ‘’are’’ utilized…
  • pp 2, line 52: ‘’proposed’’ a compositional...
  • others
  • The paper is checked for grammar errors and typos.

2-    as stated at the end of Section 2 (pp 3) the physical domain was not discussed. However, brief description could help the reader and thus is recommended.

  • A paragraph about the specifics of physical design is added.

3-    The accuracy of the classification process has been found to be enhanced by decreasing the portion of training data set (pp 7 lines 233-234. This finding may be contrary to the common practice. It is well known that the machine learning model is trained, and constructed based on the training set. That’s mean that the accuracy is expected to enhance as long as the percentage of the training data increases. Could the authors comment and justify their findings.

  • The machine learning models are checked and this error is corrected.

4- It is recommend to cite recent publication of machine learning in: Applied Sciences 11.18 (2021): 8762. DOI:https://doi.org/10.3390/app11188762

  • The recommended paper is cited in Introduction and it is under number 18 in the literature list.

Reviewer 3 Report

In this paper, a method for design of analog amplifiers based on machine learning algorithms is proposed. There are some comments should be applied, which listed as follows:

1-The introduction should be basically revised and improved. Some new papers should be added. Both parts of amplifier design and machine learning should be improved in the introduction section.

2- The heading line  “proposed method”  in line 195 should have number. Use template.

3- At least one amplifier should be designed based on this method and related curves and parameters of this amplifier should be provided.

4- This method is used to design amplifier and final goal of this method is design of an amplifier but the results of designed amplifier is not seen in the manuscript. Provide an amplifier, as a  design test, which designed with this method and explain about the schematic of PA, and related curves such as Gain, consumed power, efficiency, operating frequency and so on.

 5- Add explanation about limiting of method about operating frequency  and applications.

6- The presented results in line 250 and 261 have very low quality provide results in better format.

7- Provide comparison table and compare your method with similar works.

Author Response

Dear Reviewer, Thank you so much for your valuable comments and suggestions for the paper improvement! We are very grateful and thank you again!

In this paper, a method for design of analog amplifiers based on machine learning algorithms is proposed. There are some comments should be applied, which listed as follows:

1-The introduction should be basically revised and improved. Some new papers should be added. Both parts of amplifier design and machine learning should be improved in the introduction section.

  • The Introduction is revised and some papers are included, looking for a balance between amplifier design and machine learning topics.

2- The heading line  “proposed method”  in line 195 should have number. Use template.

  • The section Proposed method is numbered according to the template.

3- At least one amplifier should be designed based on this method and related curves and parameters of this amplifier should be provided.

  • A demonstration for designing of an amplifier with main parameters and predicted charts is added in the section Case Study.

4- This method is used to design amplifier and final goal of this method is design of an amplifier but the results of designed amplifier is not seen in the manuscript. Provide an amplifier, as a  design test, which designed with this method and explain about the schematic of PA, and related curves such as Gain, consumed power, efficiency, operating frequency and so on.

  • The design of an amplifier is demonstrated in the section Case Study.

 5- Add explanation about limiting of method about operating frequency  and applications.

  • Some explanations about the designer’s choices are added in the section Case Study and previous ones. The amplifier works in middle frequencies.

6- The presented results in line 250 and 261 have very low quality provide results in better format.

  • The pictures, taken from RapidMiner studio are replaced with text in MS Word.

7- Provide comparison table and compare your method with similar works.

  • Some related works are mentioned in the section Introduction, where the research outlines the usage of artificial neural networks to design circuits. In contrast, we apply machine learning algorithms to support the designer and to facilitate the design process.

Round 2

Reviewer 2 Report

The manuscript is recommended for the publication

Reviewer 3 Report

The authors have been addressed all of my concerns. The paper can be accepted in present form.